# Changes in the supply chain outcomes of food regionalization, 2007–2017: Broccoli in the eastern United States

**Bingyan Dai**[1]*, **Miguel I. Gómez**[1], **Shady S. Atallah**[2], **Thomas Björkman**[3]

**1** Dyson School of Applied Economics and Management, Cornell University, Ithaca, NY, United States of America, **2** Department of Agricultural and Consumer Economics, University of Illinois Urbana-Champaign, Champaign, IL, United States of America, **3** Horticulture Section, School of Integrative Plant Science, Cornell University, Ithaca, NY, United States of America

☯ These authors contributed equally to this work.
* bd393@cornell.edu

**Data Availability Statement:** All relevant data are within the paper and its Supporting Information files.

**Funding:** This work was funded by the USDA's National Institute of Food and Agriculture through the Specialty Crops Research Initiative under award

## Abstract

Local and regional food supply chains are gaining increasing support from public and private sectors for their contributions to economic development and promoting sustainability. However, the impacts of regionalization are not well understood. We employ a spatial-temporal model of production and transportation to evaluate the supply chain outcomes of a decade-long process of food regionalization for fresh broccoli in the eastern United States (US). Our results indicate that eastern broccoli supply chains displaced products sourced from the western US and met over 15% of the annual demand in eastern markets in 2017. We find that total broccoli supply chain costs and food miles increased in the period 2007–2017. Nevertheless, eastern-grown broccoli has contributed to reducing regional food miles in the eastern region (from 365 miles in 2007 to 255 miles in 2017) and experienced only modest increases in supply chains costs (a 3.4% increase, compared to a 16.5% increase for broccoli shipped from western US) during the same period. Our results provide valuable information for policymakers and the fresh produce industry interested in promoting regional food supply chains.

## Introduction

Local and regional food supply chains are essential for agricultural and rural economic development in the United States (US) [1]. They offer new marketing opportunities for farmers and food businesses in response to growing consumer demand for locally/regionally grown foods [2]. Policymakers and food businesses are increasingly supporting local and regional supply chains as a means for improving economic, environmental and social outcomes [3–5]. The US Department of Agriculture (USDA) has invested over $1 billion in more than 40,000 on local and regional food supply chain projects between 2009 and 2015 [2], and food policies have increasingly supported the expansion of these supply chains to meet growing consumer demand for local foods [6].

number 2016-51181-25402. The funders had no role in study design, data collection and analysis, decision to publish, or preparation of the manuscript.

**Competing interests:** The authors have declared that no competing interests exist.

Despite the growing importance of local and regional food supply chains, little has been done to measure their realized (or ex-post) economic and environmental impacts, particularly in fresh produce. An exception is Clancy et al. [7] which mapped regional supply chains in the northeastern US for various products including apples, cabbage, milk, and potatoes, among others. To fill this gap, we employ a spatial-temporal model of broccoli production and transportation to evaluate and compare the realized economic and environmental outcomes of a decade-long process (2007–2017) to develop the fresh broccoli supply chain in the eastern US. We solve for cost-minimizing production levels and derive seasonal transport patterns between supply and demand locations. Our analysis measures changes in supply chain costs, food miles, and the share of eastern-grown broccoli in the US eastern market between 2007–2017.

Previous literature has focused on challenges and opportunities of local/regional food supply chains and on ex-ante economic and environmental impacts based on simulations. For example, research shows that a key challenge for local/regional food supply chains is their ability to be cost competitive [8–10]. Also, earlier research using simulation (ex-ante) approaches suggests that the economic and environmental impacts of localization vary across products. For example, Atallah et al. [11] showed that increasing food regionalization may reduce total broccoli supply-chain costs and reduce food miles and the associated greenhouse gas emissions. In contrast, Nicholson et al. [12] found that the regionalization of dairy supply chains increases the total distance traveled by fluid milk with attendant increases in supply-chain costs.

Broccoli provides an excellent opportunity to assess selected economic and environmental impacts of regionalizing fresh produce supply chains. California and Arizona, which produce over 90 percent of the country's fresh broccoli [13], face long-term concerns about water availability to support water-intensive crops such as broccoli [14]. Meanwhile, new broccoli varieties suitable for production along the eastern seaboard have enabled the emergence of the broccoli supply chain in this region [11]. From 2007 to 2017, total broccoli acreage in the US eastern region increased by 60% while acres planted in the western US increased by nearly 1% (Table 1) [15, 16]. This rapid expansion provides a unique opportunity to study the evolution of a regional food supply chain over a ten-year period.

**Table 1. Estimated US Broccoli acreage from 2007 to 2017.**

| Location | 2007 base Acres | 2017 | |
| --- | --- | --- | --- |
| | | Acres | % Change |
| Maine | 5,205 | 5,948 | 14.27 |
| New York | 400 | 634 | 58.50 |
| Penn | 183 | 947 | 417.49 |
| Virginia | 551 | 411 | -25.41 |
| North Carolina | 187 | 590 | 215.51 |
| South Carolina | 750 | 886 | 18.13 |
| Georgia | 219 | 316 | 44.29 |
| Florida | 600 | 3,000 | 400.00 |
| New Jersey | 139 | 489 | 251.80 |
| Maryland | 35 | 34 | -2.86 |
| *Total Eastern* | 8,269 | 13,255 | 60.30 |
| California | 106,271 | 109,423 | 2.97 |
| Arizona | 11,869 | 9,329 | -21.40 |
| *Total western* | 118,140 | 118,752 | 0.52 |

*Source*: Estimates from eastern broccoli project director and USDA-NASS census of Agricultural in 2007 and 2017.

Our results indicate that eastern broccoli supply chains displaced products sourced from the western US and met over 15% of the annual demand in eastern markets in 2017. We find that total broccoli supply chain costs and food miles increased in the period 2007–2017. Nevertheless, eastern-grown broccoli has contributed to reducing regional food miles in the eastern region, from 365 miles in 2007 to 255 miles in 2017. Moreover, eastern-grown broccoli experienced only modest increases in supply chains costs (a 3.4% increase, compared to a 16.5% increase for broccoli shipped from western US) during the same period.

Our findings provide valuable information for policymakers and fresh produce businesses interested in promoting regional supply chains in the US. Grower-packer-shippers can benefit from the identification of supply chain configurations that can improve the economic and environmental performance of seasonal fresh produce in the easter seaboard. The methods and lessons from studying broccoli may apply to other fresh produce crops that are consumed nationally, produced consistently year-round in the western US, and potentially expandable to other regions with new varieties, while still constrained by seasonality. Examples of such specialty crops are cabbage, cauliflower, carrots, celery, chicory, endives, grapes, strawberries and lettuce, among others.

## Materials and methods

### Production and transportation model

We employ a spatial-temporal optimization model of production and transportation to compare the economic and environmental outcomes of the US eastern broccoli supply chain in 2007 and 2017. A production-transportation model allows us to analyze spatial and seasonal changes in the US broccoli supply chain, while accounting for two major characteristics: (1) fresh broccoli is perishable and seasonal, and (2) though fresh broccoli is consumed nationally, 92% of the crop is produced in California [13]. We parameterize the model with 2007 and 2017 data to solve for the production and transportation patterns that minimize total production and transportation costs in these years. Then, we compare supply-chain costs, market share and weighted average source distance (WASD) for the eastern broccoli system and conventional broccoli system and compare eastern supply chain flows in 2007 and 2017 for each season.

### Broccoli supply, demand, and transportation data

The production and transportation model requires supply, demand, and transportation data. Supply data include yield estimates, unit production costs, and land available for broccoli production at each supply location. Data pertaining to demand are seasonal volumes demanded at each regional market location. Data pertaining to transportation includes distances between supply locations and demand markets, and seasonal unit transportation costs at each supply location.

Modelled supply locations include ten fresh broccoli production regions in the eastern US, two western US producing regions (California and Arizona), and imports from Mexico and Canada (Table 2) for a total of fourteen broccoli supply regions. The fresh broccoli production regions in the eastern US are Maine, New York, Pennsylvania, Virginia, North Carolina, South Carolina, Georgia, New Jersey, Maryland, and Florida. We employ production cost estimates and yields in 2007 from recent regional broccoli crop budgets [9]. We updated yield estimates and adjusted 2017 production costs with changes in labor wages between 2007 and 2017 at each supply location using data from US Census Bureau and US Department of Labor [17, 18]. We used regional broccoli acreage estimates from USDA vegetable summary in 2007 and 2017 [15, 16] and confirm them with estimates from agricultural extension personnel in various Land Grant universities.

**Table 2. Estimated US Broccoli imports from Mexico and Canada in 2007 and 2017.**

|  | 2007 | 2017 | Change | % Change |
|---|---|---|---|---|
| Import (1,000 21lb.-boxes) |  |  |  |  |
| *Mexico* |  |  |  |  |
| Spring | 1,672 | 7,056 | 5,385 | 322 |
| Summer | 763 | 3,211 | 2,448 | 321 |
| Fall | 2,209 | 4,748 | 2,538 | 115 |
| Winter | 3,579 | 7,510 | 3,930 | 110 |
| Annual | 8,224 | 22,525 | 14,301 | 174 |
| *Canada* |  |  |  |  |
| Spring | 112 | 179 | 66 | 59 |
| Summer | 51 | 81 | 30 | 59 |
| Fall | 148 | 120 | -28 | -19 |
| Winter | 240 | 190 | -50 | -21 |
| Annual | 552 | 570 | 18 | 3 |

*Source*: Estimates from USDA-ERS Data by Commodity-Imports and Exports in 2007 and 2017.

The model has thirty-three demand nodes. We use metropolitan statistical areas (MSAs) to define demand locations in the eastern US. Demand is allocated to MSAs, and the state geographic centers based on population levels [19] and USDA's per capita disappearance for fresh broccoli in both 2007 and 2017 [20]. Supply and demand quantities are measured in 21-lb broccoli boxes. For distances between production locations and demand nodes, we use the US state spatial distance matric in [21]. We use USDA's quarterly agricultural refrigerated trucking rates [22, 23] and distances between supply locations to demand sites to calculate seasonal transportation cost for each supply location. To calculate shipment costs from Mexico, we use data from the Mexico Transport Cost Indicator Report [24, 25] and assumed the entry port is Pharr, Texas.

## Model formulation

We followed the integrated production-transportation model employed in Atallah et al. [11] and solved for optimal fresh broccoli production levels and transport patterns in 2007 and 2017 respectively, using each year's corresponding data. The model is structured as a mixed integer linear programming problem as follows:

$$\text{Minimize} \sum_{i}^{I} \sum_{k}^{K} \text{PCOST}_{i} * \text{XP}_{i,k} + \sum_{i}^{I} \sum_{j}^{J} \sum_{k}^{K} \text{TCOST}_{i,j,k} * \text{DIS}_{i,j} * \text{XT}_{i,j,k} \tag{1}$$

$$\text{Subject to :} \quad \sum_{j}^{J} \text{XT}_{i,j,k} \leq \text{XP}_{i,k} \tag{2}$$

$$\sum_{i}^{I} \text{XT}_{i,j,k} \geq \text{DEMAND}_{j,k} \tag{3}$$

$$\frac{\text{XP}_{i,k}}{\text{YIELD}_{i}} \leq \text{LAND}_{i,k} \tag{4}$$

The objective function (1) is to minimize total supply chain costs, which include total production costs for all supply locations and total transportation costs from all origins to demand locations. The model solves for two decision variables, $\text{XP}_{i,k}$ and $\text{XT}_{i,j,k}$, where $\text{XP}_{i,k}$ is the optimal production level at supply location $i$ in season $k$ and $\text{XT}_{i,j,k}$ is the optimal quantities transported from supply location $i$ to demand location $j$ in season $k$. $\text{PCOST}_{i}$ is the average total

unit production cost ($/box) in each supply location $i$ and $TCOST_{i,j,k}$ is the average total unit transportation cost ($/mile/box) from supply location $i$ to demand location $j$ in season $k$. $DIS_{i,j}$ (miles) is the distance between supply location $i$ to demand location $j$.

In order for the production and transport patterns to be feasible, they must simultaneously satisfy three constraints: (1) broccoli shipped from supply location $i$ to all demand locations $j$ in season $k$ cannot exceed production level at supply $i$ in season $k$ (Eq 2); (2) broccoli shipped to demand location $j$ in season $k$ from all supply locations $i$ have to at least satisfy the demand level ($DEMAND_{j,k}$) at each demand location $j$ in season $k$ (Eq 3); (3) land ($LAND_{i,k}$) used to produce broccoli should not exceed the available land at supply location $i$ in season $k$, where $YIELD_i$ is the supply location's average broccoli yield (Eq 4).

We use the weighted average source distance (WASD), or food miles, a measure commonly used in food system studies [26], to calculate a single distance figure that combines information on the distances from producers to consumers and the amount of product transported. We calculate eastern grown WASD (*EWASD*), mainstream sourced WASD (*MWASD*), and national WASD (*NWASD*) using the definitions in Eqs 5, 6, and 7. The calculations allow for the comparison of the food miles for two different broccoli supply chains (eastern supply chain and western supply chains) in eastern markets and to understand their changes from 2007 to 2017.

$$EWASD = \frac{\sum_i^{EORIG}\sum_j^{EDEST}\sum_k^{SEAS}DIS_{i,j}*XT_{i,j,k}}{\sum_i^{EORIG}\sum_j^{EDEST}\sum_k^{SEAS}XT_{i,j,k}} \tag{5}$$

$$MWASD = \frac{\sum_i^{MORIG}\sum_j^{EDEST}\sum_k^{SEAS}DIS_{i,j}*XT_{i,j,k}}{\sum_i^{MORIG}\sum_j^{EDEST}\sum_k^{SEAS}XT_{i,j,k}} \tag{6}$$

$$NWASD = \frac{\sum_i^{ORIG}\sum_j^{EDEST}\sum_k^{SEAS}DIS_{i,j}*XT_{i,j,k}}{\sum_i^{ORIG}\sum_j^{EDEST}\sum_k^{SEAS}XT_{i,j,k}} \tag{7}$$

*EWASD* is the average distance from eastern growers to eastern markets. *MWASD* is defined as the average distance from western growers, including California, Arizona, Mexico, and Canada, to eastern markets. *NWASD* represents the average distance from all domestic broccoli growers and imports to eastern markets. We also measure how the shares of eastern-produced broccoli and transported quantities in eastern markets change seasonally and annually in 2007 and 2017. We calculate the share of eastern-grown broccoli in East Coast markets as follows:

$$Eshare = \frac{\sum_e^{EORIG}\sum_f^{EDEST}\sum_k^{SEAS}XT_{i,j,k}}{\sum_i^{ORIG}\sum_j^{EDEST}\sum_k^{SEAS}XT_{i,j,k}} \tag{8}$$

## Results

We use fresh broccoli production levels and transportation patterns in 2007 and 2017 to calculate changes in supply chain costs, market share of eastern-grown broccoli in eastern markets, and food miles for the western- and eastern-grown broccoli in eastern markets in the US. We also report seasonal eastern supply chain flows for 2007 and 2017 (Fig 1 to Fig 8).

### Changes in the supply-chain costs

Results indicate that, in 2007, the landed costs (i.e., the costs of producing and transporting broccoli to eastern US demand locations) of a 21 lb. -box of broccoli produced in the East

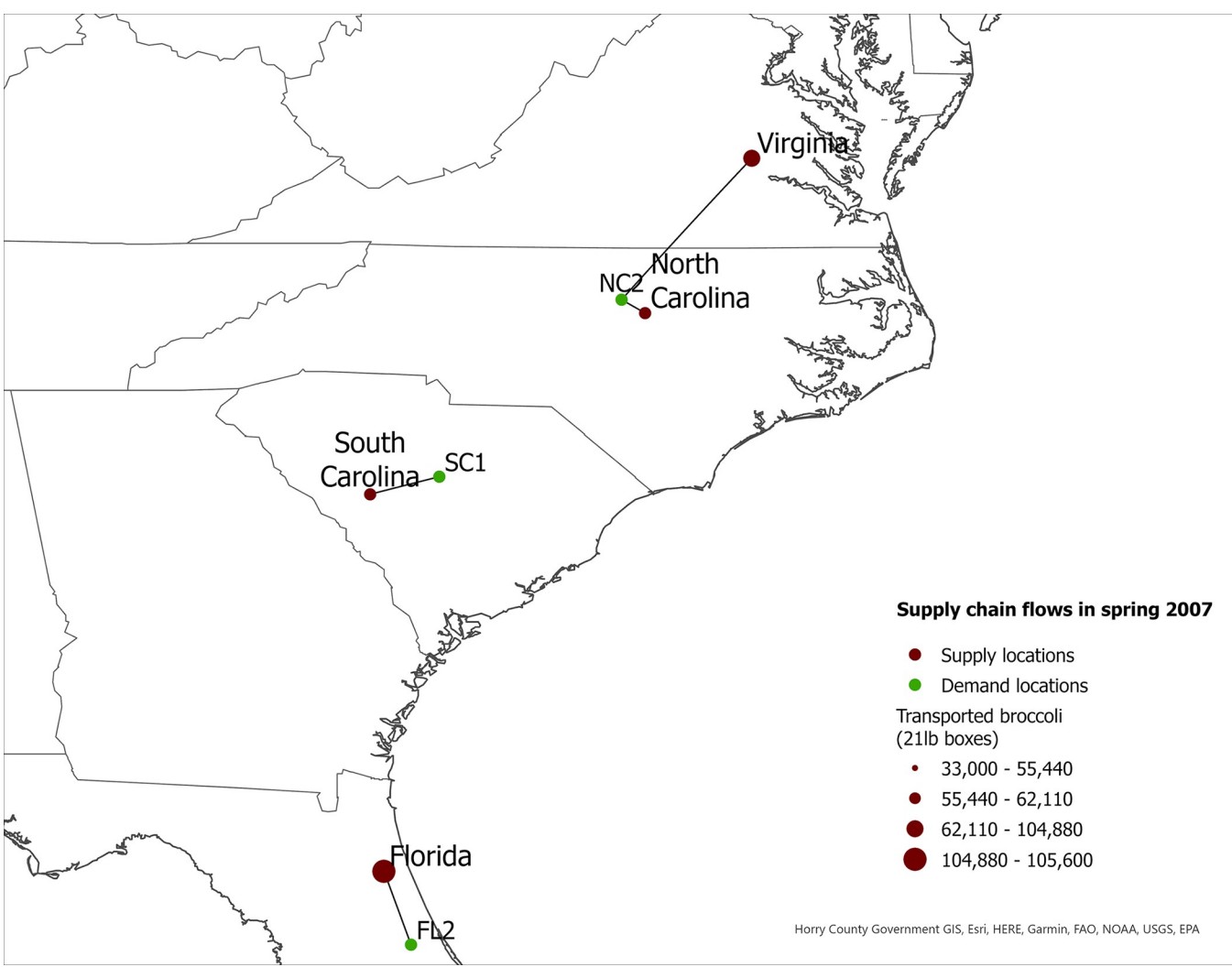

**Fig 1. US eastern broccoli supply chain flows in spring 2007.**

Coast were higher than a box of broccoli sources from the West Coast (Table 3). However, the per box cost of eastern-grown broccoli increased only modestly between 2007 and 2017, and was lower than the landed cost of a box of broccoli sourced from the West Coast in 2017. The landed costs for eastern-grown broccoli increased from $13.90/box to $14.37/box between 2007 and 2017, a 3% increase. In contrast, the landed cost of a box of broccoli sourced from the West Coast increased from $13.36/box to $15.57/box in the same period, a 17% increase. The average supply-chain costs of fresh broccoli to meet demand in the eastern US, which combines broccoli from all eastern region states and mainstream regions (West Coast and imports), increased by 14% in 2017 (Table 3).

The supply-chain costs of eastern-grown broccoli in 2007 varied across different seasons. During winter, the supply-chain costs for eastern-grown broccoli are the lowest at approximately $12 per box. Florida was the only supply location producing broccoli in winter 2017 on the East Coast and it was solely meeting local demand. Eastern-grown broccoli is more costly per box in the summer and fall season, at around $14 per box. As in 2007, the supply-chain costs in 2017 varied across different seasons, with an increase in costs observed in all seasons.

**Table 3. Changes on supply-chain costs in US eastern markets.**

|  | 2007 | 2017 | Change | % Change |
|---|---|---|---|---|
| *Supply-chain costs ($/box)* |  |  |  |  |
| Average supply-chain costs of all sourced broccoli |  |  |  |  |
| Spring | 14.22 | 15.19 | 0.97 | 7 |
| Summer | 14.32 | 15.69 | 1.37 | 10 |
| Fall | 13.00 | 15.53 | 2.53 | 19 |
| Winter | 12.26 | 15.16 | 2.90 | 24 |
| Annual | 13.45 | 15.39 | 1.94 | 14 |
| Supply-chain costs of eastern-grown broccoli |  |  |  |  |
| Spring | 13.35 | 13.66 | 0.31 | 2 |
| Summer | 14.02 | 14.81 | 0.79 | 6 |
| Fall | 14.06 | 14.88 | 0.83 | 6 |
| Winter | 11.99 | 13.06 | 1.07 | 9 |
| Annual | 13.90 | 14.37 | 0.48 | 3 |
| Supply-chain costs of mainstream sourced broccoli |  |  |  |  |
| Spring | 14.14 | 15.37 | 1.23 | 9 |
| Summer | 14.39 | 15.86 | 1.47 | 10 |
| Fall | 12.57 | 15.75 | 3.18 | 25 |
| Winter | 12.27 | 15.36 | 3.10 | 25 |
| Annual | 13.36 | 15.57 | 2.21 | 17 |

*Source*: Authors' calculations based on the optimization model.

As more land acreage was allocated to broccoli production in eastern US in 2017, eastern-grown broccoli was transported to more local and regional markets on the East Coast, which has contributed to the increased costs. The supply-chain costs of broccoli sourced from western US and imports were lower in the fall and winter seasons than in the spring and summer seasons in 2007. However, in 2017, these seasonal differences in supply chain costs for mainstream-sourced broccoli were almost non-existent.

## Changes in the share of eastern-grown broccoli in the eastern US

Our results indicate that with a 60% increase of broccoli acreage in eastern regions between 2007 and 2017, the annual share of eastern-grown broccoli to meet demand in this region increased from 12% to 15% in this period (Table 4). Eastern-grown broccoli gained market share primarily in spring and winter, 6.5 and 6.7 percentage points, respectively. In contrast, the market share of eastern-grown broccoli decreased slightly in the summer and fall seasons (0.6 and 2.8 percentage points, respectively). The amount of broccoli shipped from eastern suppliers to East Coast demand locations increased substantially over the ten-year span. The largest change occurred in the seasons with increased market share (spring and winter). Winter shipments were up by 435% and spring shipments were up 232% (Table 4). Shipments within the eastern regions increased even in the seasons in which market share declined. Summer shipments were up by 21% and fall shipments were up 14% (Table 4).

## Changes in eastern broccoli supply chain flows, 2007–2017

The expansion of broccoli production in the eastern US is associated with the spatial reorganization of the broccoli supply chain. In this section we present our findings regarding product flow changes in the eastern US between 2007–2017 for each season. The changes in product flows are presented in Table 5 and in Figs 1–8.

**Table 4. Changes on broccoli shipments to meet eastern US demand, 2007–2017.**

|  | 2007 | 2017 | Change | % Change |
|---|---|---|---|---|
| Market share of eastern-grown broccoli (%) |  |  |  |  |
| Spring | 4.00 | 10.53 | 6.53 | n/a |
| Summer | 16.36 | 15.73 | -0.63 | n/a |
| Fall | 28.87 | 26.03 | -2.84 | n/a |
| Winter | 2.05 | 8.70 | 6.65 | n/a |
| Annual | 12.48 | 15.02 | 2.54 | n/a |
| Transported eastern-grown broccoli quantities (1,000 21lb.-boxes) |  |  |  |  |
| Spring | 306 | 1,016 | 710 | 232 |
| Summer | 1,237 | 1,500 | 264 | 21 |
| Fall | 2,041 | 2,323 | 282 | 14 |
| Winter | 158 | 847 | 689 | 435 |
| Annual | 3,741 | 5,686 | 1,945 | 52 |

*Source*: Authors' calculations based on the optimization model.

**Table 5. Changes on seasonal broccoli flows (1,000 21lb.-boxes) from eastern locations and mainstream (West Coast and import) locations to eastern markets in 2007 and 2017.**

| Demand locations | 2007 | | 2017 | |
|---|---|---|---|---|
|  | Eastern supply | Mainstream supply | Eastern supply | Mainstream supply |
| *Spring* |  |  |  |  |
| Charlotte, North Carolina | 0 | 246 | 104 | 222 |
| Charleston, South Carolina | 0 | 84 | 120 | 0 |
| Augusta, Georgia | 0 | 73 | 61 | 34 |
| Gainesville, Florida | 0 | 226 | 55 | 239 |
| *Summer* |  |  |  |  |
| Buffalo, New York | 90 | 164 | 143 | 158 |
| New York city, New York | 146 | 1,366 | 184 | 1,660 |
| Charlotte, North Carolina | 16 | 227 | 52 | 270 |
| Bridgeport, Connecticut | 142 | 0 | 82 | 90 |
| Hartford, Connecticut | 119 | 0 | 128 | 13 |
| *Fall* |  |  |  |  |
| Buffalo, New York | 90 | 148 | 143 | 138 |
| Albany, New York | 0 | 115 | 86 | 53 |
| New York city, New York | 823 | 592 | 549 | 1,176 |
| Charlotte, North Carolina | 0 | 227 | 104 | 197 |
| Charleston, South Carolina | 0 | 78 | 56 | 54 |
| Durham, North Carolina | 242 | 23 | 137 | 212 |
| Gainesville, Florida | 0 | 209 | 28 | 245 |
| *Winter* |  |  |  |  |
| Augusta, Georgia | 0 | 74 | 55 | 41 |
| Deltona, Florida | 0 | 598 | 792 | 216 |

*Source*: Authors' calculations based on the optimization model.

## Spring season

According to the model, supply chain flows from eastern supply locations to eastern markets increased substantially between 2007 and 2017 (Figs 1 and 2). In spring 2007, 100% of broccoli shipped to Charlotte, NC came from the West Coast or was imported (Table 5). In contrast, by 2017, 32% of the broccoli shipped to this demand location came from the eastern US (Table 5). Another example of the increased importance of eastern-grown broccoli is the Charleston, SC demand location: In spring 2007, 100% of the broccoli shipped to Charleston, SC was sourced from outside the region; but by spring 2017, 100% of the broccoli shipped to this demand location was sourced from the eastern US. Similar shifts in sourcing occurred in Augusta, GA and Gainesville, FL. In 2007, both demand locations relied on western suppliers for 100% of their broccoli. However, spring 2017, eastern-grown broccoli represented 64% and 19% of total demand in Augusta and Gainesville, respectively.

## Summer season

In contrast to the spring season, our results suggest modest product flow changes between 2007 and 2017 (Figs 3 and 4), with a few exceptions. The proportion of eastern-grown broccoli

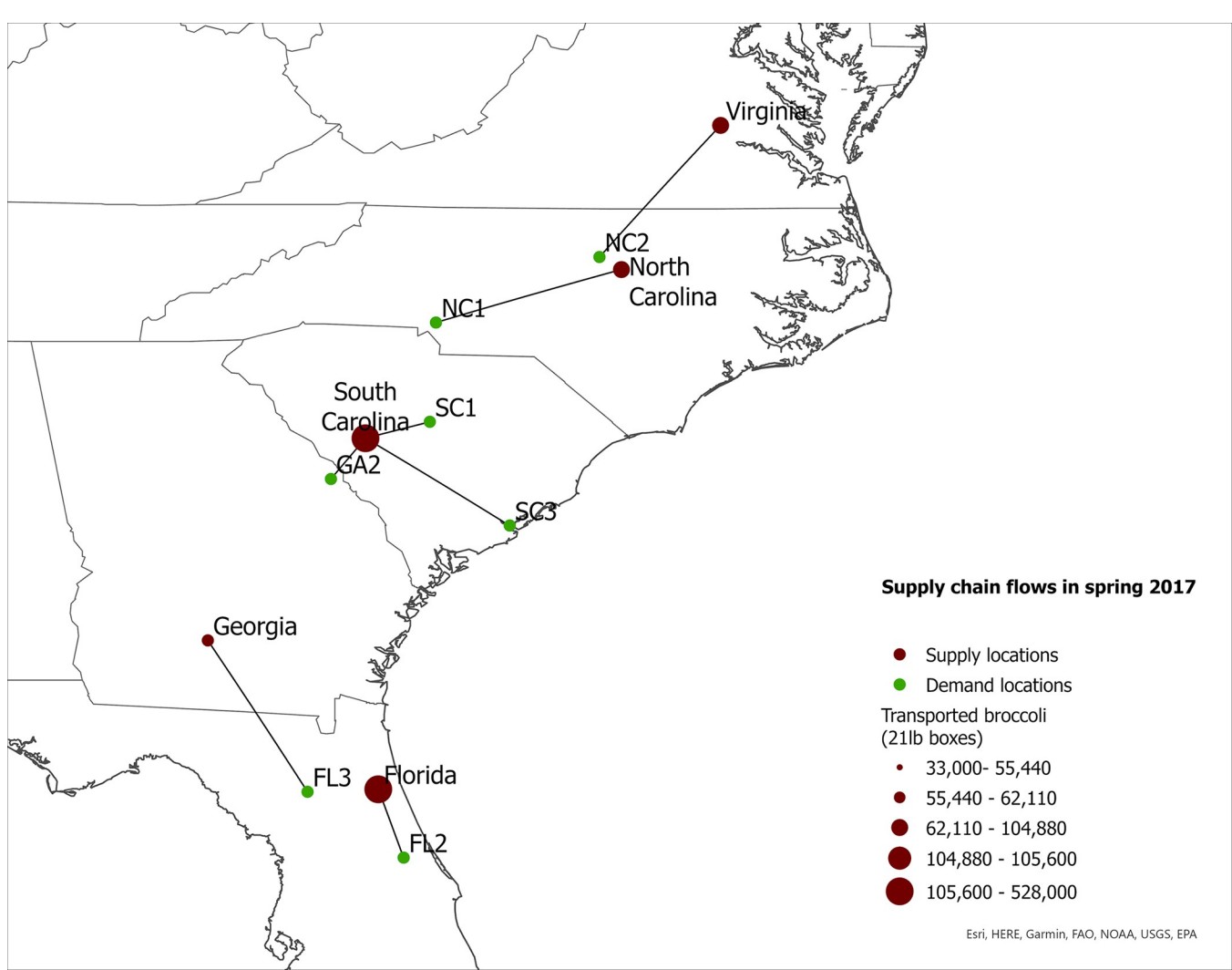

**Fig 2. US eastern broccoli supply chain flows in spring 2017.**

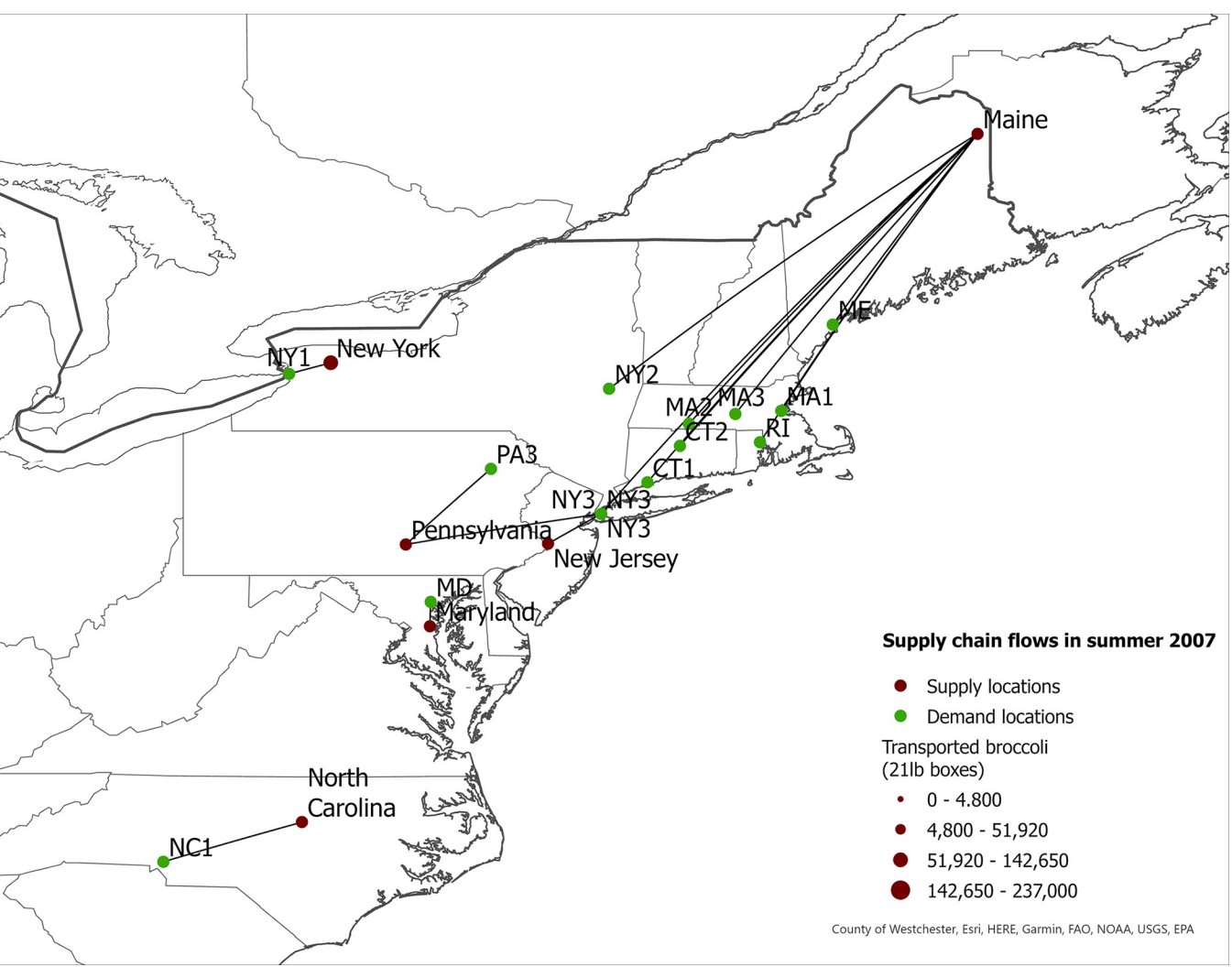

**Fig 3. US eastern broccoli supply chain flows in summer 2007.**

shipped to the Buffalo, NY demand location (NY1) increased from 35% in 2007 to 48% in 2017, an increase of 13 percentage points over the ten-year period. Conversely, in both Bridgeport, CT and Hartford, CT sourced 100% broccoli from eastern supply regions during summer 2007. However, by 2017, Bridgeport, CT (CT1) and Hartford, CT (CT2) sourced 52% and 9% of the broccoli from elsewhere. These two locations sourced their eastern-grown broccoli from Maine in both years. The results indicate substantial increases of broccoli grown in Maine and shipped to eastern demand locations from 2007 to 2017. Maine shipped more broccoli to other demand locations (MA 1, MA2, MA3, ME) with a shorter shipping distance. As a result, CT1 and CT2 increased sourcing from mainstream regions. These flow changes illustrate that not all demand locations in the East Coast increased regional sourcing.

**Fall season.** Figs 5 and 6 present the optimal broccoli flows in 2007 and 2017. Georgia was not a supplier of broccoli in 2007, but by 2017 it became a supplier to multiple MSAs in Florida. Other increases in eastern broccoli shipments to demand location in the region occurred in New York, North Carolina, and South Carolina. In 2007, all broccoli shipped in the fall to Albany, NY (NY2), Charlotte, NC (NC1), Charleston, SC (SC3) and Gainesville, FL (FL3)

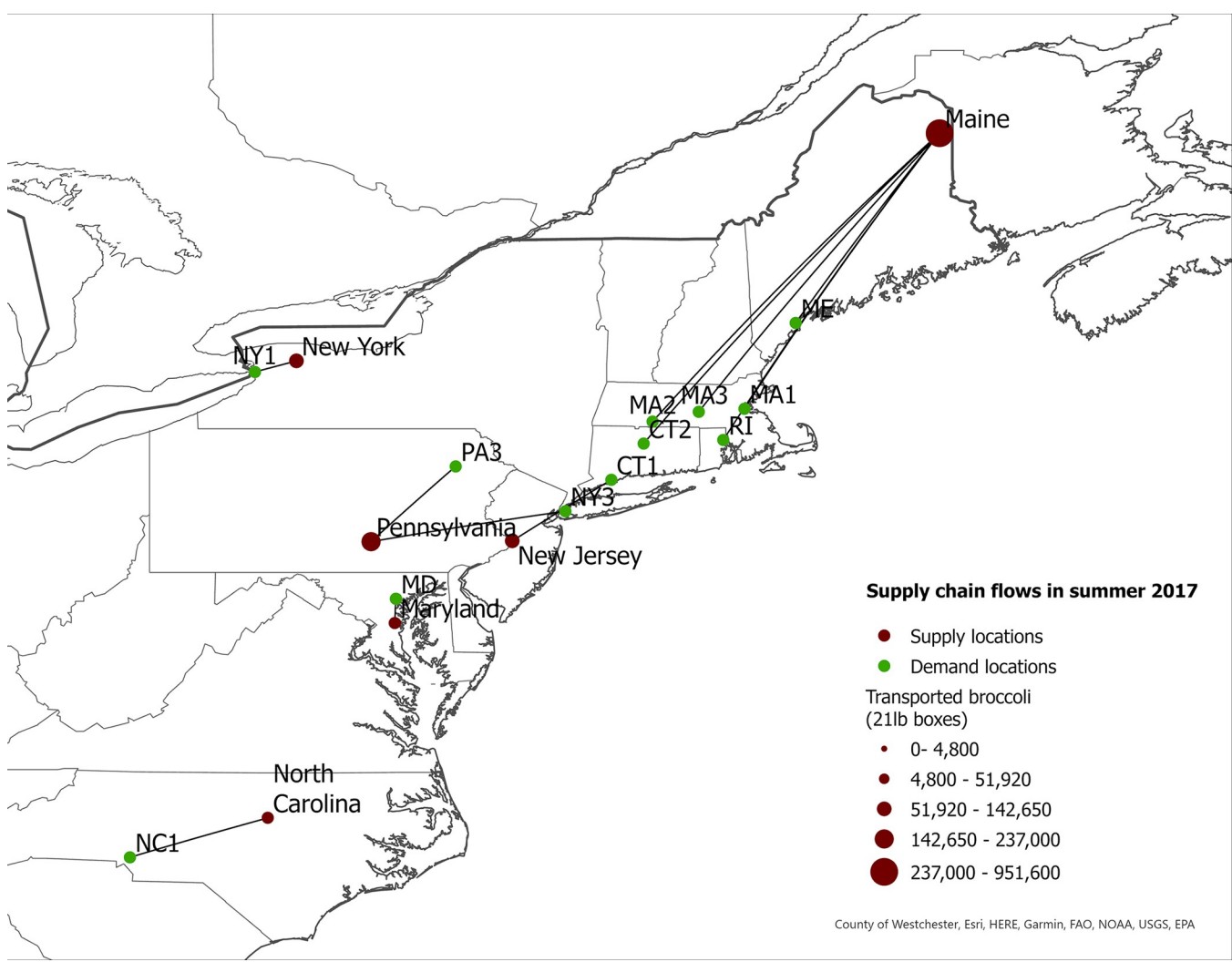

**Fig 4. US eastern broccoli supply chain flows in summer 2017.**

originated in mainstream supply locations (Figs 5 and 6). By 2017, however, these four demand locations sourced the majority of their broccoli from eastern states. For example, in 2017, 62% and 51% of broccoli demanded was from the eastern region in NY1 and SC3, respectively (Table 5). Nevertheless, some demand locations increased sourcing from outside the East Coast. For example, in New York City (NY3) and Durham, NC (NC2), the shares of eastern-grown broccoli decreased by 2017 resulting in over 50% of supplied broccoli to both cities coming from mainstream regions. Maine was the main broccoli supplier to eastern states for NY3 in 2007, but broccoli from this state was shipped to other demand locations located closer to production sites than New York City. Similarly, North Carolina and Virginia supplied 91% of broccoli demanded in Durham, NC; but by 2017, 60% of shipments came from California. These results suggest that not all demand locations have seen increases in eastern-grown broccoli in the fall season.

**Winter season.** In winter 2017, broccoli supply locations expanded to include Georgia (supplying to demand locations in the same state) in addition to Florida (Figs 7 and 8). Results also indicate broccoli quantities shipped from Florida and Georgia to eastern markets also

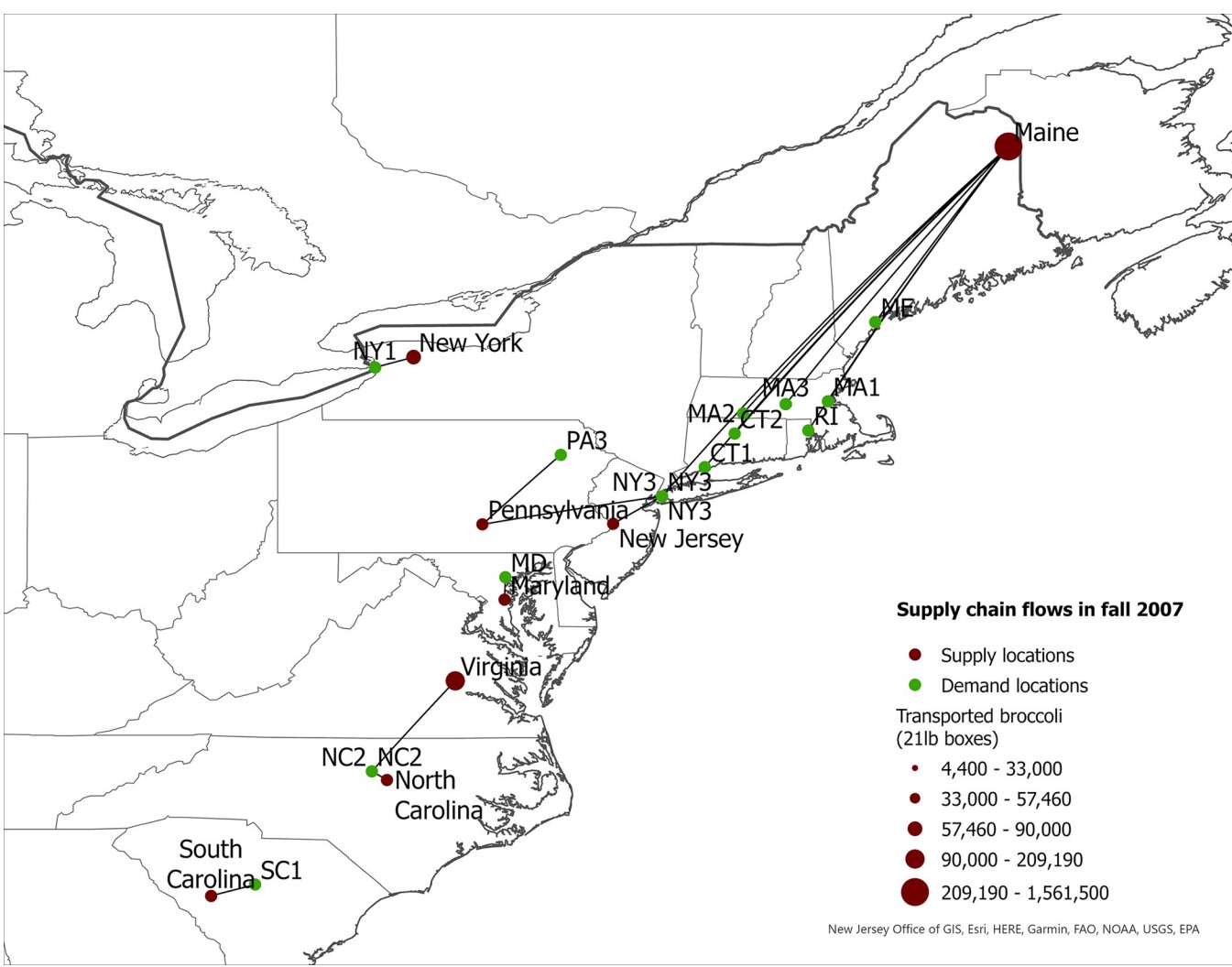

**Fig 5. US eastern broccoli supply chain flows in fall 2007.**

increased substantially from 2002 to 2017. In 2007, 100% of broccoli demand in Augusta, GA (GA2) and Deltona, FL (FL2) was satisfied by mainstream regions (Table 5). By 2017, eastern-grown broccoli provided 58% and 79% of broccoli in Augusta (GA2) and Deltona (FL2), respectively.

## Changes in food miles

The regional broccoli supply chains in eastern markets meet the expectation that the product travels less distances than broccoli coming from elsewhere. The largest broccoli production states, Maine and Florida, are about 1,600 miles apart. This distance is smaller than the one that separates Atlanta, GA and Salinas, CA (around 2,400 miles) or the one that separates it from Guanajuato, Mexico (around 1,650 miles). Atlanta, GA, as an example of a smaller market and long distances from the West Coast, makes a strong case that the difference in food miles needs to be quantified. The annual average distance travelled from eastern broccoli growers to eastern markets (365 miles) was far less than that from the out-of-region supply (2,533 miles; Table 6). The annual average distance from eastern broccoli regions to eastern

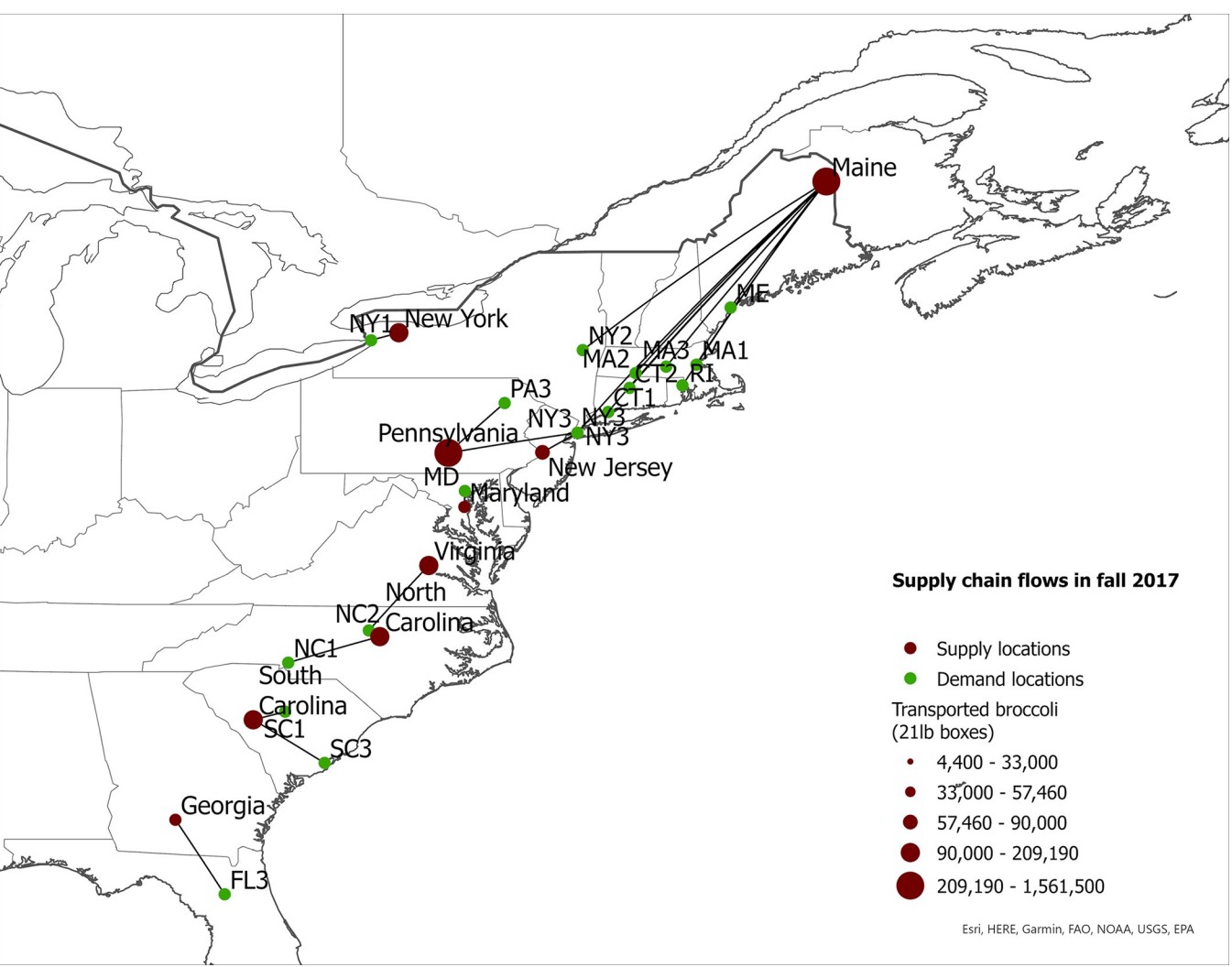

**Fig 6. US eastern broccoli supply chain flows in fall 2017.**

markets decreased by 30%, from 365 miles in 2007 to 255 miles in 2017 (Table 6). This decrease is due to changes in product flows whereby eastern produced broccoli substituted some of the mainstream produced broccoli in eastern states. In contrast, the annual average distance of broccoli produced in mainstream source locations increased by 11%, from 2,533 miles in 2007 to 2,817 miles in 2017.

The average distance travelled from eastern broccoli growers to eastern markets exhibited seasonal variation in both 2007 and 2017. During the summer and fall, eastern-grown broccoli travelled longer distances compared to the spring and winter seasons, due to increased shipment volumes to more local and regional areas. In 2017, the average distance travelled from eastern broccoli growers to eastern markets decreased slightly during summer and fall seasons, while it increased during spring and winter due to changes in market share (Table 4). In addition, the average distance by broccoli sourced in mainstream locations also displayed seasonal differences. During the summer season, mainstream-sourced broccoli traveled the farthest distance (2,726 miles), while the distance travelled was relatively shorter during the fall and winter seasons (2,373 miles and 2391 miles, respectively). Notably, in 2017, mainstream-sourced

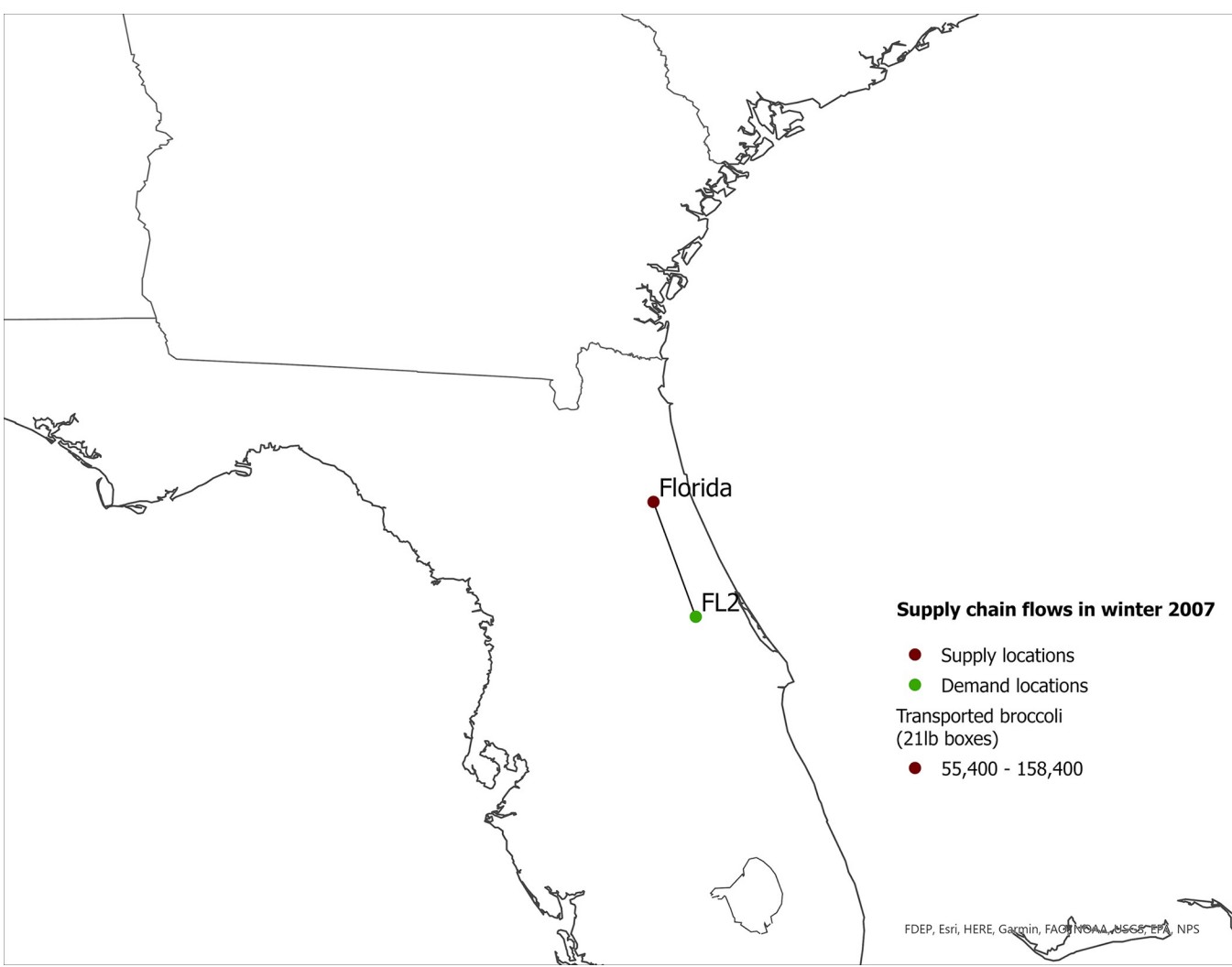

**Fig 7. US eastern broccoli supply chain flows in winter 2007.**

broccoli travelled a longer distance during winter, exceeding the farthest distance travelled in summer 2007.

## Discussion

The model solutions provide several insights regarding how the regionalization of the broccoli supply chains evolved over ten years (2007–2017) and its impacts on supply chain costs, market share, product flows and food miles. First, the model results show that eastern broccoli supply chains met over 15% of the annual demand in eastern markets in 2017, with modest increases in supply-chain costs. Despite eastern states having limited land with an appropriate climate to produce broccoli in spring and winter, market shares of eastern grown broccoli in these two seasons have increased by more than 6 percentage points from 2007 to 2017. One important advantage of sourcing eastern grown broccoli is the lower supply chain cost compared to the broccoli sourced from the mainstream system in 2017. Supply-chain costs of western-sourced broccoli are likely to rise due to increased land rent and limited irrigation water in western production regions due to rationing in times of drought. This is likely to increase

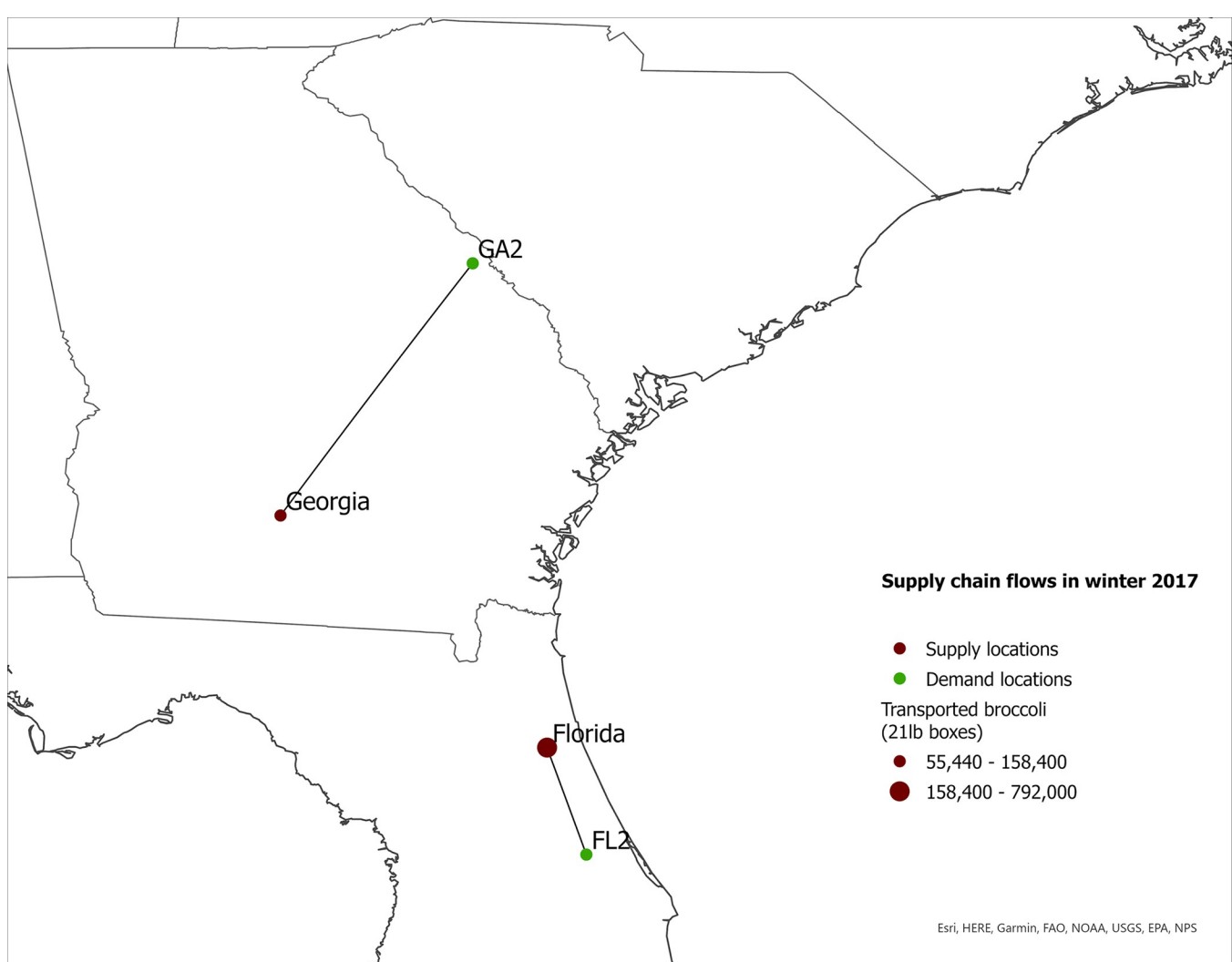

**Fig 8. US eastern broccoli supply chain flows in winter 2017.**

incentives to further develop broccoli supply chains on the East Coast to attend this region's demand.

Second, the broccoli supply chain became shorter in 2017 compared with 2007, and its structure changed depending on demand sites and seasons. Not surprisingly, the average distance travelled from mainstream supply sites including the West Coast, Mexico, and Canada to eastern markets is much higher (2,817 miles in 2017), about eleven times as far as the distance travelled from eastern locations (255 miles in 2017). Thousands of miles saved could produce massive reductions in carbon emissions and pollution. Moreover, some supply chain flows in the eastern broccoli system changed unexpectedly. Changes in the share of eastern sourced broccoli and mainstream sourced broccoli supply for eastern demand locations differ across locations and seasons. The optimal supply chain structure we identified in this paper might provide insights for local and regional policies when designing desired distribution networks.

This analysis also helps inform questions on the relationship between the mainstream and regional food supply chains. Our results demonstrate that the eastern broccoli supply chain

**Table 6. Changes on weighted average source distance (WASD) travelled from different supply systems in US eastern markets in 2007 and 2017.**

|  | 2007 | 2017 | Change | % Change |
|---|---|---|---|---|
| *WASD (miles)* |  |  |  |  |
| WASD of all national sourced broccoli (*NWASD*) |  |  |  |  |
| Spring | 2,520 | 2,559 | 39 | 2 |
| Summer | 2,342 | 2,403 | 61 | 3 |
| Fall | 1,806 | 2,141 | 335 | 19 |
| Winter | 2,345 | 2,602 | 257 | 11 |
| Annual | 2,262 | 2,432 | 171 | 8 |
| WASD of eastern sourced broccoli (*EWASD*) |  |  |  |  |
| Spring | 111 | 130 | 20 | 18 |
| Summer | 383 | 287 | -96 | -25 |
| Fall | 408 | 324 | -84 | -21 |
| Winter | 158 | 160 | 2 | 1 |
| Annual | 365 | 255 | -110 | -30 |
| WASD of mainstream sourced broccoli (*MWASD*) |  |  |  |  |
| Spring | 2,625 | 2,845 | 220 | 8 |
| Summer | 2,726 | 2,798 | 73 | 3 |
| Fall | 2,373 | 2,780 | 407 | 17 |
| Winter | 2,391 | 2,834 | 444 | 19 |
| Annual | 2,533 | 2,817 | 284 | 11 |

*Source*: Authors' calculations based on the optimization model.

has become integral to the entire national broccoli supply by complementing broccoli sourced from mainstream regions. Our cost-minimizing supply chain solution includes both mainstream and eastern locations shipping product to meet demand in the East Coast. Our results show that the eastern broccoli system has lower supply-chain costs and food miles, which will accumulate and become significant over time. As the dominant supplier, however, the mainstream broccoli system still accounts for high national market shares. Policymakers and businesses should focus on how to leverage the advantages of developing the eastern broccoli supply chain and its potential contribution to improve economic and environmental of the national supply chains. Policymakers should consider the impacts on the mainstream food system when designing policies to expand local and regional food systems. The model solutions might provide insights for structuring the future optimal supply system by giving the optimal magnitude, locations, and seasons. However, since the food system is more complex and outcomes such as social and environmental impacts should be taken into account to assess the performance of regional supply chains for fresh produce in general.

## Conclusions

In this paper we employed a spatial-temporal model of production and transportation of the US broccoli sector to analyze changes and impacts of the evolving eastern broccoli supply chains from 2007 to 2017. The primary conclusion is that the eastern broccoli supply chain has gained market share during this period and has contributed to lower costs and food miles for broccoli consumers in the region. The increased regionalization in the period 2007–2017 modestly increased the total supply chain costs, moderating the substantial increases in production costs in supply locations outside the region, specifically in the West Coast. In addition, the food miles of broccoli produced in the east coast decreased substantially between 2007 and

2017 (from 265 miles to 255 miles per box). These positive trends and increasing market size also present opportunities for growers and food businesses to expand regional food production. Although the model is used to assess the US eastern broccoli systems, we believe that our model is generalizable and can be adopted to evaluate and track other regional supply chains for fresh produce.

Our estimated decade-long economic and environmental effects of regionalization of the broccoli supply chains offers insights into the optimal planning and design of local and regional supply chains for policy makers and other stakeholders. Product flow reorganizations occur differently for each season and vary across demand locations in the Eastern US. Our results suggest that the eastern broccoli supply chain shows modest changes in summer and fall, while major flow changes occurred in spring and winter. Therefore, grower-packer-shippers and policymakers should carefully consider opportunities to regionalize supply chains, because the impacts vary by season, and location.

Our findings are valuable to the fresh produce industry, policy makers and other stakeholders interested in promoting localization of supply chains. Nonetheless, our model has several limitations that deserve further investigation. First, fresh produce supply chains might be influenced by several supply disruptions, such as the water crisis in western US. Long-lasting droughts are driving the water shortages across much of the west and are likely to reduce crop yields, lead farmers to plant fewer acres and increase production costs for irrigation water, to reduce acreage, or to abandon production altogether. The spatial-temporal model can be expanded to evaluate the impacts of such disruptions and provide valuable information on optimal locations to replace production regions affected by disruptions. Such changes might increase broccoli production costs in traditional production areas on the West Coast. Researchers can use our transshipment model to identify the optimal eastern supply locations to meet national demand while minimizing the total production and transportation costs. Second, we assumed that broccoli acreage increases in the eastern states are a small fraction of total fresh vegetable acreages. Future research can include the crops that compete with broccoli for land in each eastern state and the opportunity costs of broccoli acreage expansion in the model. Third, our model does not consider East Coast consumer preferences for broccoli produced in the region. Future research can examine consumer preferences for regionally produced broccoli and the implications for economic and environmental outcomes. Finally, we focus on a specific commodity (broccoli) to illustrate the impacts of localizing fresh produce supply chains. Future studies can adapt our model to examine the impacts of regionalizing supply chains for other fresh produce sectors.

## Supporting information

**S1 Dataset.**
(XLSX)

## Author Contributions

**Conceptualization:** Bingyan Dai, Miguel I. Gómez.

**Data curation:** Bingyan Dai, Thomas Björkman.

**Formal analysis:** Bingyan Dai, Miguel I. Gómez.

**Funding acquisition:** Miguel I. Gómez, Thomas Björkman.

**Investigation:** Bingyan Dai, Miguel I. Gómez.

**Methodology:** Bingyan Dai, Miguel I. Gómez, Shady S. Atallah.

**Project administration:** Bingyan Dai, Miguel I. Gómez.

**Resources:** Miguel I. Gómez.

**Software:** Bingyan Dai, Miguel I. Gómez, Shady S. Atallah.

**Supervision:** Miguel I. Gómez.

**Validation:** Bingyan Dai, Miguel I. Gómez, Shady S. Atallah, Thomas Björkman.

**Visualization:** Bingyan Dai.

**Writing – original draft:** Bingyan Dai, Miguel I. Gómez, Shady S. Atallah, Thomas Björkman.

**Writing – review & editing:** Bingyan Dai, Miguel I. Gómez, Shady S. Atallah, Thomas Björkman.

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
