## [Decision Letter · Decision Letter 0]

27 Mar 2023

PONE-D-23-03462A decade later: Changes in the supply chain outcomes of food regionalization, 2007-2017: Broccoli in the eastern United StatesPLOS ONE

Dear Dr. Dai,

Thank you for submitting your manuscript to PLOS ONE. After careful consideration, we feel that it has merit but does not fully meet PLOS ONE’s publication criteria as it currently stands. Therefore, we invite you to submit a revised version of the manuscript that addresses the points raised during the review process.

We look forward to receiving your revised manuscript.

Kind regards,

Ibrahim Badi, PhD

Academic Editor

PLOS ONE

Journal Requirements:

3. We note that All Figures in your submission contain [map/satellite] images which may be copyrighted. All PLOS content is published under the Creative Commons Attribution License (CC BY 4.0), which means that the manuscript, images, and Supporting Information files will be freely available online, and any third party is permitted to access, download, copy, distribute, and use these materials in any way, even commercially, with proper attribution. For these reasons, we cannot publish previously copyrighted maps or satellite images created using proprietary data, such as Google software (Google Maps, Street View, and Earth). For more information, see our copyright guidelines: http://journals.plos.org/plosone/s/licenses-and-copyright.

a. You may seek permission from the original copyright holder of All Figures to publish the content specifically under the CC BY 4.0 license.  

4. Please upload a copy of Supporting Information Figure/Table/etc. Figures 1-8 which you refer to in your text on page 29.

Reviewers' comments:

Reviewer's Responses to Questions

**Comments to the Author**

1. Is the manuscript technically sound, and do the data support the conclusions?

Reviewer #1: Partly

Reviewer #2: Yes

2. Has the statistical analysis been performed appropriately and rigorously? 

Reviewer #1: Yes

Reviewer #2: Yes

3. Have the authors made all data underlying the findings in their manuscript fully available?

Reviewer #1: Yes

Reviewer #2: Yes

4. Is the manuscript presented in an intelligible fashion and written in standard English?

Reviewer #1: Yes

Reviewer #2: Yes

5. Review Comments to the Author

Reviewer #1: The article is topical. However, the following revisions may further improve the clarity

a. The abstract and Introduction section are long.

b. Please present the result section in a lucid way

c. Add few more future directions

Reviewer #2: The paper evaluates the supply chain outcomes of a decade-long process of food regionalization for fresh broccoli in the United States and found that eastern broccoli supply chains displaced product sourced in the western US, meeting over 15% of the annual demand in eastern markets in 2017. However, the results regarding supply-chain costs and food miles were not consistent with previous research which predicted that increasing localization of fresh broccoli would decrease total supply-chain costs and food miles in eastern markets.

6. PLOS authors have the option to publish the peer review history of their article (what does this mean?). If published, this will include your full peer review and any attached files.

Reviewer #1: No

Reviewer #2: No

---

## [Author Response · Author response to Decision Letter 0]

15 May 2023

May 11th, 2023 

Dr. Ibrahim Badi

Academic Editor

PLOS ONE

Re: Re-submission Manuscript PONE-D-23-03462 for PLOS ONE

Dear Dr. Badi, 

Thank you for the opportunity to revise and resubmit our manuscript entitled “Changes in the supply chain outcomes of food regionalization, 2007-2017: Broccoli in the eastern United States” and for the thoughtful comments provided by you and the reviewers. We appreciate the time and effort that went into the review process. We have carefully considered the feedback provided by you and the reviewers and have made significant revisions to the manuscript.

Reviewer 1 provided valuable feedback on the length and clarity of our abstract and introduction, as well as the presentation of our results. In response, we made substantial changes in most sections of the manuscript including the Abstract and the Introduction, Results, Discussion and Conclusion sections to make them clearer and more concise. We have also expanded future directions for research in the Conclusion section. 

Reviewer 2 provided a valuable comment regarding our interpretation of results regarding supply chains costs and food miles. In response, we addressed his/her concerns in the Discussion and Conclusion.

Please see details of the changes we made in the manuscript in the response to each reviewer below. 

Regarding journal requirements, we have made the necessary changes to our manuscript in accordance with your request. Specifically: 

• In our Data Availability statement, in the original submission we did not specify where the minimal data set underlying the results described in our manuscript can be found. In response, we are now providing our data set, which will be included as supporting information files. 

• Regarding the copyright of map images in the original submission, we were unable to obtain permission from the original copyright holder. In response, we now provide replacement figures that comply with the CC BY 4.0 License. We used ArcGIS to create all the figures. Using ArcGIS maps in academic publications is permitted by the terms of use for ArcGIS static maps (https://doc.arcgis.com/en/arcgis-online/reference/static-maps.htm). We uploaded a new copy of supporting information, Figures 1-8 which we refer to in our text on page 29. 

• We have conducted a thorough review of our manuscript and can confirm that it adheres to PLOS ONE’s style requirements. Furthermore, we have carefully revised our reference list and can verify that it is complete and correct. 

My co-authors and I believe that the revisions we have made have substantially improved the quality of our manuscript. We have taken into consideration all of the comments and feedback provided by you and the reviewers and have carefully revised the manuscript accordingly. To help with the review process, we have attached both a marked-up copy that highlights the changes made to the original version and an unmarked version of our revised paper without track changes. We hope that this will assist in evaluating the changes that we have made.

Thank you again for the opportunity to revise our manuscript. I look forward to hearing back from you soon.

Sincerely,

Bingyan Dai 

 

Response to Reviewer # 1

Manuscript PONE-D-23-03462 “Changes in the supply chain outcomes of food regionalization, 2007-2017: Broccoli in the eastern United States”

Overview of response: Thank you for taking the time to review our manuscript and provide us with valuable and constructive feedback. Your comments helped us improve the quality of our manuscript, and we are grateful for your comments. In response to your comments, we revised the Abstract and the Introduction section to make them more concise, to the point, and to improve the clarity of exposition. In addition, we rewrote substantial portions of the Results section to clearly present our findings. Furthermore, in the Conclusion section, we now discuss limitations of the study and suggest areas for future research. You will find your original comments in bold and our response right below each comment.

The article is topical. However, the following revisions may further improve the clarity

a. The abstract and Introduction section are long.

We agree with the reviewer. In response, we completely rewrote the Abstract and the Introduction section to clearly state the relevance of our research question, our approach to answering the research question, and the findings of our study. In doing so, we substantially shortened the Abstract and the Introduction section to improve the clarity of communication and the flow of the manuscript. Please see the new Abstract and the new Introduction section (pages 2-5).

b. Please present the result section in a lucid way.

Thank you for this comment. We agree that the Results section needs to be substantially improved. In response, we practically rewrote this section to clearly discuss our finding; to present more relevant detail on the cost, flows and food mile changes between 2007 and 2017; and to clearly and precisely interpret the model results. Please see the revised Results section (pages 10-19).

c. Add few more future directions.

This is an excellent suggestion. In response, we elaborated more on the discussion of future research directions. Specifically, we now discuss four areas for future research as follows (see new paragraph in page 22): 

1) Fresh produce supply chains might be influenced by several supply disruptions, such as the water crisis in western US. Long-lasting droughts are driving the water shortages across much of the west and are likely to reduce crop yields, lead farmers to plant fewer acres and increase production costs for irrigation water, to reduce acreage, or to abandon production altogether. The spatial-temporal model can be expanded to evaluate the impacts of such disruptions and provide valuable information on optimal locations to replace production regions affected by disruptions. Such changes might increase broccoli production costs in traditional production areas on the West Coast. Researchers can use our transshipment model to identify the optimal eastern supply locations to meet national demand while minimizing the total production and transportation costs. 

2) We assumed that broccoli acreage increases in the eastern states are a small fraction of total fresh vegetable acreages. Future research can include the crops that compete with broccoli for land in each eastern state and the opportunity costs of broccoli acreage expansion in the model. 

3) Our model does not consider East Coast consumer preferences for broccoli produced in the region. Future research can examine consumer preferences for regionally produced broccoli and the implications for economic and environmental outcomes. 

4) We focus on a specific commodity (broccoli) to illustrate the impacts of localizing fresh produce supply chains. Future studies can adapt our model to examine the impacts of regionalizing supply chains for other fresh produce sectors.

 

Response to Reviewer # 2

Manuscript PONE-D-23-03462 “Changes in the supply chain outcomes of food regionalization, 2007-2017: Broccoli in the eastern United States”

Overview of response: Thank you for taking the time to review our manuscript and provide us with valuable and constructive feedback. In response to your comment and the comments from other reviewer, we revised the Abstract and the Introduction section to make them more concise, to the point, and to improve the clarity of the exposition. In addition, we rewrote substantial portions of the Results section to clearly present our findings and clarify the results regarding costs and food miles. Furthermore, in the Conclusion section, we now discuss limitations of the study and suggest areas for future research. You will find your original comment in bold and our response right below each comment.

The paper evaluates the supply chain outcomes of a decade-long process of food regionalization for fresh broccoli in the United States and found that eastern broccoli supply chains displaced product sourced in the western US, meeting over 15% of the annual demand in eastern markets in 2017. However, the results regarding supply-chain costs and food miles were not consistent with previous research which predicted that increasing localization of fresh broccoli would decrease total supply-chain costs and food miles in eastern markets.

Thank you for this comment. The reviewer is correct that the results regarding supply-chain costs and food miles were not consistent with previous research which predicted that increasing localization of fresh broccoli would decrease total supply-chain costs and food miles in eastern markets. 

The national supply chain costs increased because the cost of west coast-grown broccoli increased substantially (from 13.36 $/box in 2007 to 15.57 $/box in 2017). However, we note that the costs of east coast-grown broccoli increased only modestly (from 13.90 $/box in 2007 to 14.37 $/box in 2017) (see Table 3). Therefore, we argue that east coast-grown broccoli has contributed to avoiding substantial cost increases for broccoli in this region. We now explain this clearly in the revised Discussion and in the Conclusion sections.

Regarding changes in food miles, we agree that our discussion of results in the original submission was not clear. National food miles increased because the distances traveled by broccoli originated outside the east coast increased (from 2,533 miles in 2007 to 2,817 miles in 2017). Nevertheless, the distance traveled by east coast-grown broccoli decreased (from 365 miles in 2007 to 255 miles in 2017) (see Table 6). Therefore, east coast-grown broccoli has contributed to moderate the increase in food miles of the national supply chain shipping broccoli to the east coast region. We now explain this clearly in the revised Discussion and in the Conclusion sections.

---

## [Decision Letter · Decision Letter 1]

12 Jun 2023

Changes in the supply chain outcomes of food regionalization, 2007-2017: Broccoli in the eastern United States

PONE-D-23-03462R1

Dear Dr. Dai,

We’re pleased to inform you that your manuscript has been judged scientifically suitable for publication and will be formally accepted for publication once it meets all outstanding technical requirements.

Kind regards,

Ibrahim Badi, PhD

Academic Editor

PLOS ONE

Additional Editor Comments (optional):

Reviewers' comments:

Reviewer's Responses to Questions

**Comments to the Author**

1. If the authors have adequately addressed your comments raised in a previous round of review and you feel that this manuscript is now acceptable for publication, you may indicate that here to bypass the “Comments to the Author” section, enter your conflict of interest statement in the “Confidential to Editor” section, and submit your "Accept" recommendation.

Reviewer #1: All comments have been addressed

2. Is the manuscript technically sound, and do the data support the conclusions?

Reviewer #1: Yes

3. Has the statistical analysis been performed appropriately and rigorously? 

Reviewer #1: Yes

4. Have the authors made all data underlying the findings in their manuscript fully available?

Reviewer #1: Yes

5. Is the manuscript presented in an intelligible fashion and written in standard English?

Reviewer #1: Yes

6. Review Comments to the Author

Reviewer #1: I don't have any further comment. The authors have addressed the concerns raised earlier. The authors are requested for checking minor typos

7. PLOS authors have the option to publish the peer review history of their article (what does this mean?). If published, this will include your full peer review and any attached files.

Reviewer #1: No

---

## [Editor Report · Acceptance letter]

21 Jun 2023

PONE-D-23-03462R1 

Changes in the supply chain outcomes of food regionalization, 2007-2017: Broccoli in the eastern United States 

Dear Dr. Dai:

I'm pleased to inform you that your manuscript has been deemed suitable for publication in PLOS ONE. Congratulations! Your manuscript is now with our production department. 

Kind regards, 

on behalf of

Dr. Ibrahim Badi 

Academic Editor

PLOS ONE